# Geometric and topological characterization of the cytoarchitecture of islets of Langerhans

**Manu Aggarwal**[1][⊕]*, **Deborah A. Striegel**[1][⊕], **Manami Hara**[2], **Vipul Periwal**[1]

**1** Laboratory of Biological Modeling, NIDDK, National Institutes of Health, Bethesda, Maryland, United States of America, **2** Section of Endocrinology, Diabetes and Metabolism, Department of Medicine, The University of Chicago, Chicago, Illinois, United States of America

⊕ These authors contributed equally to this work.
\* manu.aggarwal@nih.gov

**Data Availability Statement:** All data and code is made available at https://github.com/nihcompmed/Pancreatic-Islets.

**Funding:** This research was supported by the Intramural Research Program of the NIH, the National Institute of Diabetes and Digestive and

## Abstract

The islets of Langerhans are critical endocrine micro-organs that secrete hormones regulating energy metabolism in animals. Insulin and glucagon, secreted by beta and alpha cells, respectively, are responsible for metabolic switching between fat and glucose utilization. Dysfunction in their secretion and/or counter-regulatory influence leads to diabetes. Debate in the field centers on the cytoarchitecture of islets, as the signaling that governs hormonal secretion depends on structural and functional factors, including electrical connectivity, innervation, vascularization, and physical proximity. Much effort has therefore been devoted to elucidating which architectural features are significant for function and how derangements in these features are correlated or causative for dysfunction, especially using quantitative network science or graph theory characterizations. Here, we ask if there are non-local features in islet cytoarchitecture, going beyond standard network statistics, that are relevant to islet function. An example is ring structures, or cycles, of $\alpha$ and $\delta$ cells surrounding $\beta$ cell clusters or the opposite, $\beta$ cells surrounding $\alpha$ and $\delta$ cells. These could appear in two-dimensional islet section images if a sphere consisting of one cell type surrounds a cluster of another cell type. To address these issues, we developed two independent computational approaches, geometric and topological, for such characterizations. For the latter, we introduce an application of topological data analysis to determine locations of topological features that are biologically significant. We show that both approaches, applied to a large collection of islet sections, are in complete agreement in the context both of developmental and diabetes-related changes in islet characteristics. The topological approach can be applied to three-dimensional imaging data for islets as well.

## Author summary

The pancreatic islets or islets of Langerhans are regions of the pancreas that contain endocrine or hormone-producing cells classified as alpha, beta, delta, PP, and epsilon. They are responsible for regulating blood glucose levels, and their dysfunction leads to diabetes. Differences in relative arrangement of alpha, beta, and delta cells has been observed between species. For example, mouse islets predominantly contain beta cells in the central

Kidney Diseases (NIDDK) (ZIA DK075038-09: VP and DS; ZIA DK075161-02: VP and MA). The funders had no role in study design, data collection and analysis, decision to publish, or preparation of the manuscript.

**Competing interests:** The authors have declared that no competing interests exist.

core with alpha and delta cells forming a mantle around them localized in the periphery. Similar arrangement has been seen in small human islets, but larger islets have alpha and delta cells also along the vessels penetrating inside the islet. These findings motivate the debate of the functional significance of these structural patterns. In this work we mathematically define and implement quantitation of ring-structures or cycles of alpha and delta cells around beta cells (and vice versa) using two distinct methods, geometric and topological, in 2D sections of islets. We analyze two different data sets of 2D sections of human islets, one from different developmental stages and another from control and diabetic subjects. Further, we illustrate extension of the topological method to three dimensional data sets.

## Introduction

Described by Paul Langerhans as part of his medical dissertation in 1869, the islets of Langerhans [1] are endocrine micro-organs embedded in the acinar tissue of the exocrine pancreas. While they comprise only about 1–4% of the total mass of the pancreas [2], they produce and secrete hormones that are crucial for regulating blood glucose levels as well as levels of amino acids, free fatty acids, keto acids, glycerol, and other energy-rich nutrients. There are several cell types within islets, including beta cells, alpha cells, delta cells, and others, each of which produces a specific hormone with complex counter-regulatory actions. These cells communicate through a complex network of paracrine [3] and autocrine signaling pathways involving hormones, neuropeptides, growth factors, and through electrical coupling via gap junctions [4, 5].

Beta cells are responsible for producing insulin, which promotes the uptake and storage of glucose in muscle and adipose tissue and inhibits hepatic glucose production. Insulin also affects the storage of glucose in the liver in the form of glycogen, as well as promotes storage of lipids in fat tissue (inhibition of the hormone-sensitive lipase, promotion of lipoprotein lipase), and amino acids in muscle tissue. Alpha cells produce glucagon, which increases blood glucose levels by promoting the breakdown of glycogen stored in the liver and muscles, and by stimulating gluconeogenesis, the production of glucose from non-carbohydrate sources such as amino acids and fatty acids. Delta cells produce somatostatin, which helps regulate the secretion of both insulin and glucagon. Electrical connectivity of delta cells is just being appreciated in recent research and they play a central role in the regulation of both insulin and glucagon [6]. Other cell types in the islets, such as epsilon cells, produce hormones that are involved in appetite regulation and the overall metabolic response to food intake. These cells are not randomly arranged in the islets of Langerhans, but instead form a complex three-dimensional architecture. It is believed that islets have a core of beta cells ($\beta$), surrounded by mantles of alpha cells and delta cells ($\alpha\delta$) that are close to the periphery, with the outer layer consisting of pancreatic polypeptide (PP) cells. For example, [7] observed that small human islets (40–60$\mu$m in diameter) in their study had $\beta$-cells in a core position, $\alpha$-cells in a mantle position, and vessels at the periphery. In bigger islets they observed $\alpha$-cells in a mantle position as well as along vessels that penetrated and branched inside the islets. There is some degree of variation in the precise arrangement of the cells between islets, which themselves come in a variety of sizes, and between species [2, 8, 9]. The precise arrangement of cells may play a role in the signaling between cells and the regulation of hormone secretion [7, 10–13]. How this arrangement of different cell types in islets [14] relates to innervation [15] of, and blood flow [16, 17] through islets is still an area of active study [18, 19]. Motivated by the debate about the functional

significance of core-mantle segregation of islet cells, our aim in this work is to quantitate ring-structures or cycles of $\alpha\delta$-cells that surround $\beta$-cells and cycles of $\beta$-cells that surround $\alpha\delta$-cells in 2D and 3D data sets of islet cell composition.

The development of islet cell populations and their numbers during the transition from birth to adulthood is also a complex process that is not fully characterized in humans. However, studies have shown that there are dynamic changes in the numbers of different cell types during this period [20, 21]. For example, the number of beta cells in the islets increases significantly during the first few months after birth, then gradually increases until the age of about 5–6 years, after which it remains relatively stable until early adulthood. The numbers of other islet cell types, such as alpha and delta cells, also change during this period, although the patterns and magnitudes of these changes may differ from those of beta cells. There can be significant individual variability in the numbers of islet cells during development, and environmental factors such as diet and lifestyle also influence these processes. The relationship between islets and acinar and ductal cells may also be different at different developmental stages and in different species (e.g., islets are more intralobular in humans and more interlobular in mice).

In type 2 diabetes, it has been possible to quantitatively assess the decrease in the number and function of different cell types in the islets of Langerhans [18, 22, 23]. Beta cell mass and insulin secretion decrease, and delta cell number and somatostatin secretion also decrease. Early during the development of diabetes there may be an adaptive increase in mass or function (or both) and in some people (non-progressors), this may be more functional or persistent than in others (progressors) and is crucial for our understanding why some people do not develop T2D despite insulin resistance and why some respond to dietary interventions while others do not [24–29]. Glucagon secretion is inhibited by insulin but is also regulated by a complex interplay of several other factors, including glucose levels, amino acids, and neural inputs. Counterintuitively, the regulation of glucagon secretion is impaired, leading to an increase in glucagon secretion. The mechanisms underlying this are not fully understood [30–32], but it may be due to a combination of factors besides impaired insulin secretion, including decreased sensitivity of alpha cells to glucose and altered gut hormone signaling. In addition, inflammation and oxidative stress [33], which are known to be elevated in diabetes, can also affect alpha cell function and contribute to increased glucagon secretion. This progressive loss of cellular function ultimately leads to impaired glucose homeostasis and hyperglycemia in type 2 diabetes.

Quantitative study of the arrangement of islet cells in health, disease and development has been largely defined by the availability of data, both imaging and electro-physiological. The majority of imaging data consists of immuno-fluorescence in two-dimensional (2D) sections of islets which are then processed with image analysis software to determine nuclear locations and cell types. Patch-clamp electro-physiological studies measure the electrical activity of individual cells within isolated islets, and are used to investigate the functional connectivity of the islets by recording the activity of cells in response to different stimuli [34–38]. Calcium imaging using fluorescent dyes monitors changes in intracellular calcium levels in response to different stimuli and allows the measurement of the coordinated activity of cells within the islet [39].

Given this data, the complex architecture of the islets of Langerhans has been described quantitatively using a variety of methods. In the context of islets of Langerhans, structural connectivity refers to the physical connections between different cell types within the islet, such as gap junctions [35, 40] that facilitate direct communication and signaling between the cells. Functional connectivity [36] refers to the coordination and synchronization of different cell types within the islet. This is essential for proper regulation of glucose metabolism and insulin

secretion. Spatial statistics, such as spatial auto-correlation functions which describe the degree of similarity between nearby cells, describe the distribution and arrangement of cells within islets quantitatively [19]. Network science [18, 21, 41, 42], going beyond spatial statistics by computing centrality measures, clustering coefficients, and the modularity of the graph defined by the cell-to-cell adjacency matrix obtained from images, has also been used to find quantitative characteristics of islet morphology as evidenced in 2D sections. New imaging techniques have been used to create detailed three-dimensional (3D) images of islets, which have been used to quantify relationships between intraislet capillary density and islet size [43].

More recently, there has been significant activity combining network science with mathematical modeling to understand how alterations in structural and/or functional connectivity affect islet response to glucose-stimulated insulin secretion. The role of hub nodes in networks [44–49] has been central to this activity with some conflicting reports. In this context, we set out to investigate if there are quantitative topological characteristics distinct from the network statistics that have been investigated for islet networks that may be relevant for dynamical islet function. These nonlocal topological characteristics may, for example, implicitly incorporate information about innervation and vascularization. They are also specific to the dimensionality of the islet data, unlike functional network characteristics which can in principle be defined without using any information about the spatial milieu of islet cells.

In this work, first, we develop a geometric quantitative characterization of the occurrence of ring structures or cycles comprising of $\beta$ cells ($\beta$-cycles) or $\alpha\delta$ cells ($\alpha\delta$-cycles) that surround cells of the other type(s) in 2D islet sections. We noted that this geometric approach was unlikely to generalize robustly to 3D islet images that are becoming available. Therefore, to check the results obtained via this intricate geometric quantitation, we developed a topological data analysis approach that could be checked to agree with the geometric 2D results, and at the same time generalize to 3D data. Specifically, persistent homology (PH) is a branch of topological data analysis that computes topological invariants from networks by varying a minimal edge-length threshold in a network. This allows PH to uncover non-local robust network features. A 2D or 3D data set of locations of cells in an islet is a network of cells spatially embedded in a Euclidean space. In this case, features computed by PH can be geometrically realized as holes in the spatial embedding. Hence, $\alpha\delta$-cycles of interest can be computed as representative boundaries of holes in the embedded network of $\alpha\delta$-cells that contain $\beta$-cells inside. However, representative boundaries around these features are not unique by definition [50] so the computed boundary may be geometrically imprecise. In [51] we developed technical tools for finding tight representatives for topological features that improve geometrical precision of estimation of their location. Here, we introduce a way to quantitate topological features that are biologically significant, specifically, those features in the spatial embedding of the $\alpha\delta$-network that contain $\beta$-cells inside and vice versa. With these tools in hand, we have investigated a large number of islet sections in this paper in an attempt to approach the cytoarchitecture of the islets of Langerhans in a computationally rigorous setting. We applied our computational approach to both developmental changes in islet cytoarchitecture and to compare diabetic and control islets.

## Results

Two data sets of locations of beta ($\beta$), alpha ($\alpha$), and delta ($\delta$) cells in 2D slices of human pancreatic islets were analyzed. The first comprises islets in different developmental stages of gestation (stage 0), 1–35 weeks (stage 1), 12–24 months (stage 2), and 28 months and later (stage 3). Changes in pancreatic islet cyto-architecture during development have been studied previously using this data set [21]. The second comprises islets from diabetic and non-diabetic

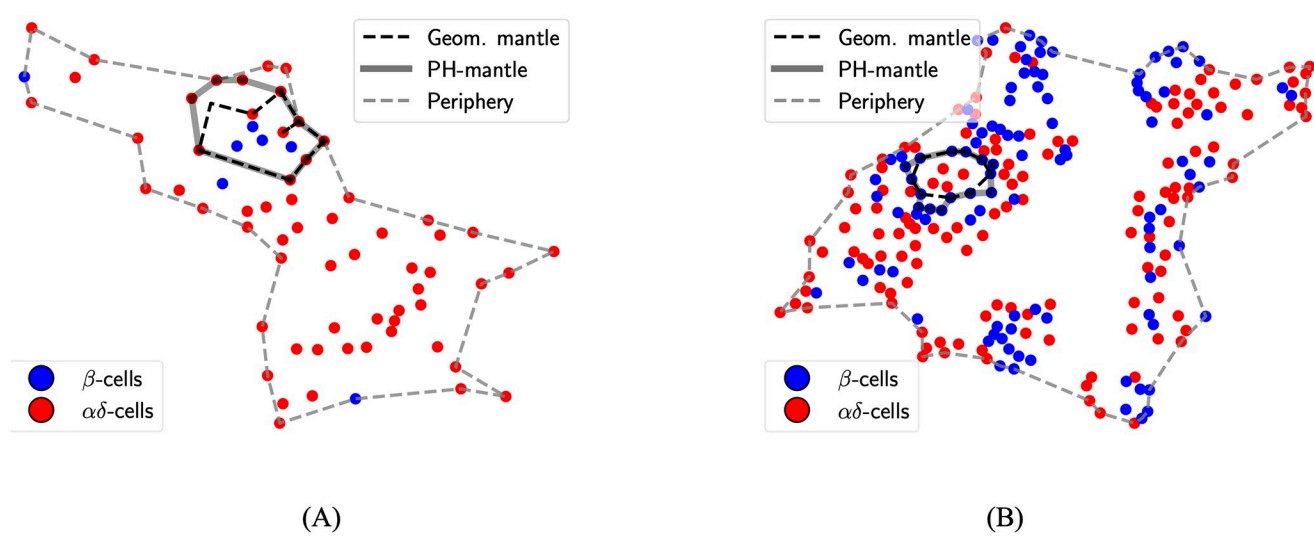

(A)                                    (B)

**Fig 1. Examples of geometric and PH-cycles.** (A) $\alpha\delta$-cycle around a NS $\beta$-component. (B) $\beta$-cycle around a NS $\alpha\delta$-component.

human subjects. We call the former developmental data set and the latter the T2D data set. Subject-wise demographics for the T2D data sets are shown in S4 Table. These details are not available for the developmental data set. $\alpha$ and $\delta$ cells together will be denoted by $\alpha\delta$-cells. Ring structures computed using the geometrical method will be called geometric cycles and those computed using topological data analysis will be called PH-cycles. Fig 1 shows examples of cycles around non-singular (NS) components (component of a network with more than one node or cell in this case that were computed using both methods.

## Cell composition of islets changes significantly during development

There were 6088, 4942, 3130, and 7203 islets with at least five $\beta$-cells and five $\alpha\delta$-cells in stages 0 to 3, respectively. Islets were characterized by their total number of cells (transformed to log scale) and $\beta$-cell fraction. The resulting 2D distributions of islet characteristics for each stage were compared between stages pairwise using the Kolmogorov–Smirnov test (KS-test). We found that the distributions of islet characteristics are significantly different between every pair of stages ($p$-values $< 0.05$, see S1 Table). For a more informative comparison, we used the Kullback–Leibler divergence (KL-divergence) to quantitatively assess the relative difference between kernel density estimates (KDEs) of the 2D distributions of islet characteristics. Fig 2A shows that the KL-div between stages 0 and 1 and between 2 and 3 are smaller than all other pairwise comparisons. This indicates that islet cell composition changes more significantly from stage 1 to stage 2. Peaks of the KDEs indicate that a higher proportion of islets have higher $\beta$-cell fraction in the later developmental stages of 2 and 3. S10A Fig shows exemplary 2D sections (characteristics similar to the peak of the KDEs) for the developmental stages. We next plot the KDE of the 2D distribution of islets characterized by the number of $\alpha$ and $\delta$ cells in them. Only islets with at least 5 $\alpha$ and $\delta$-cells were considered. Fig 2B shows that in the early stages (0 and 1) majority of islets have the same number of $\alpha$ and $\delta$-cells. However, in the later stages of development clusters of islets appear that have more $\alpha$ than $\delta$-cells. Moreover, almost all of the control and diabetic islets (with at least 5 cells of each kind) have more $\alpha$ than $\delta$-cells (see S9 Fig). We note that control and diabetic islets are from older human subjects (see S4

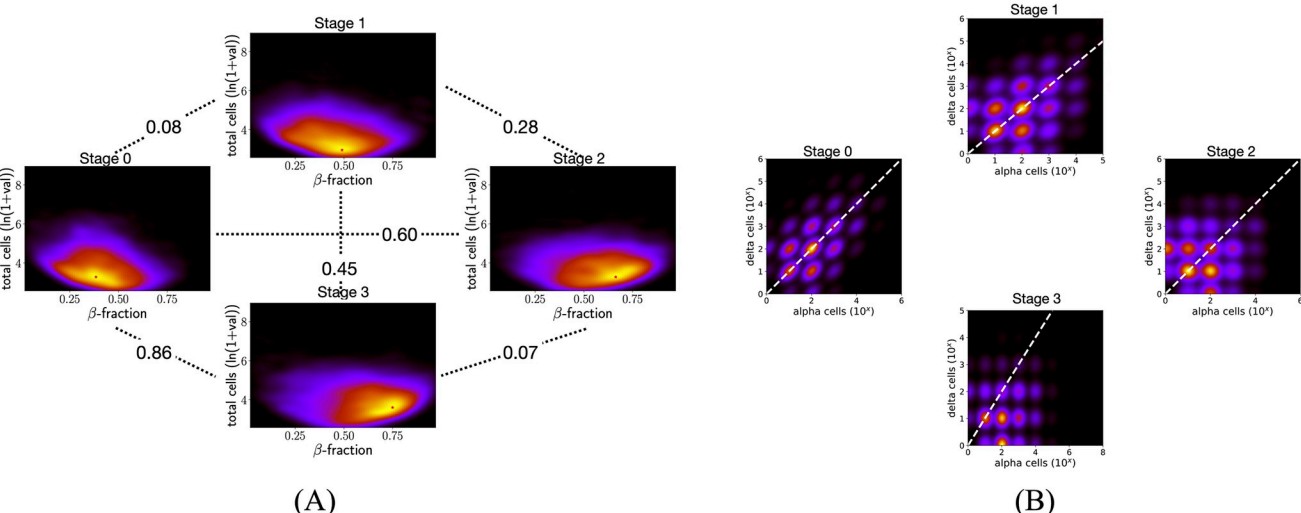

**Fig 2. Comparing cell composition across developmental stages.** (A) KDE plots of the distribution of islets characterized by the total number of cells and $\beta$-cell fraction from stage 0 to stage 3. The number between every pair of plots shows the KL-divergence between respective KDEs. The KL-divergence between stages 0 and 1 and between 2 and 3 are at most 0.07 as compared to at least 0.27 for every other pairwise comparison. The mode of the density estimate is marked by a red star in each plot. Peaks are at (0.38, 3.27), (0.49, 2.95), (0.66, 3.27), and (0.75, 3.6) for stages 0 to 3. A higher proportion of islets in the last two stages have a higher $\beta$-cell fraction. (B) KDE plots of distribution of islets characterized by number of $\alpha$ and $\delta$-cells. In the later developmental stages (2 and 3) there is a large proportion of islets with more $\alpha$-cells than $\delta$-cells (bright regions in the KDE under the $y = x$ white dashed line).

Table). The distribution of ages of all subjects in the T2D data set has a minimum age of 15 years and the median age of 64 years.

## There is a correlation between changes in cycle formation around cores and topology of the islets across developmental stages

Computing geometric cycles found 649, 463, 168, and 453 islets with at least one NS $\beta$-component inside an $\alpha\delta$-cycle in stages 0 to 3, respectively. In contrast, there were 64, 88, 238, and 823 islets with at least one NS $\alpha\delta$-component inside a $\beta$-cycle in stages 0 to 3, respectively. Fig 3A plots the percentages with respect to the total number of islets. Albeit these percentages are small, the trends in percentages of islets with $\alpha\delta$-cycles around $\beta$-cores and $\beta$-cores around $\alpha\delta$-cycles are clearly different, with significant changes in both happening after stage 1. Further, a similar pattern is observed in maximum dimension-1 ($H_1$) topological persistence of all islets. S1 Appendix gives an illustration of dimension-1 persistence applied to a point-cloud and an intuitive interpretation of results. S8 Fig shows distributions of the maximum dimension-1 persistence for $\beta$-graphs (left panel) and $\alpha\delta$-graphs (right panel) in all 2D sections across developmental stages. Except for stages 0 and 1 all other pairwise comparisons (Mann-Whitney U test) show that the distributions are significantly different. Fig 3B shows the median and 95%-tile of distributions of maximum $H_1$ of $\alpha\delta$-cells and of $\beta$-cells in all islets at different developmental stages. Both the median and 95%-tile of $\beta$-cells increase after stage 1, suggesting that there are holes with larger robustness that can wrap around NS $\alpha\delta$-components. Similarly, both the median and 95%-tile of the maximum $H_1$ persistence of $\alpha\delta$-cells decrease from stage 1 to 2 in accordance with a decrease in the percentage of islets with at least one NS $\beta$-component inside an $\alpha\delta$-cycle. S1 Fig shows that distributions of maximum persistence are significantly different after stage 1.

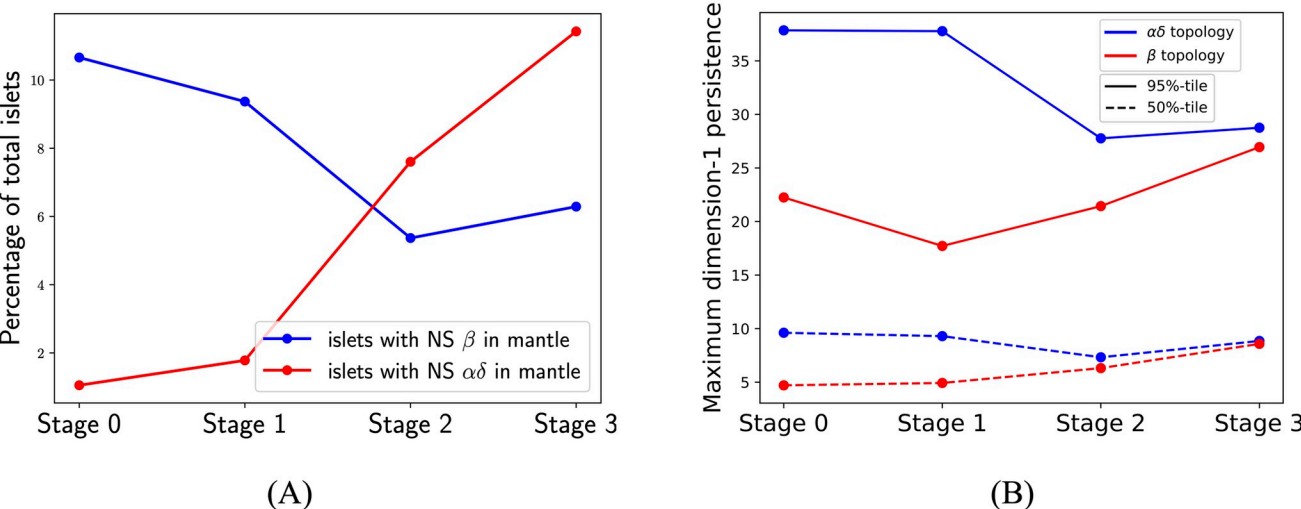

**Fig 3. Changes in cycle around core formation during development correlate with changes in the topology of islets.** (A) Percentages of islets with at least one NS $\beta$-component in a $\alpha\delta$-cycle (blue) and those with at least one NS $\alpha\delta$-component in a $\beta$-cycle (red). The trends of the two are distinct. (B) Percentiles of distributions of maximum persistences of $\alpha\delta$-cells (blue) and $\beta$-cells (red) in islets. Trends in (B) correlate with those in (A) from stage 1 onwards.

## Islets with at least one NS $\beta$-cell component inside $\alpha\delta$-cycles have differences in their cell composition between later stages

Fig 2A showed that the KL-divergence in the distribution of characteristics of all islets between stages 2 and 3 is the smallest ($\approx 0.07$) amongst all pairwise comparisons. Further, the percentages of islets with a NS $\beta$-component inside an $\alpha\delta$-cycle are similar in stages 2 and 3 (see Fig 3A). However, Fig 4A shows that distributions of characteristics of such islets have a larger KL-divergence of 0.26 between stages 2 and 3. Specifically, a comparison of peaks of these KDEs indicates that a higher proportion of such islets in stage 3 contain more cells. In contrast, Fig 4B shows that islets with a NS $\alpha\delta$-component inside a $\beta$-cycle have smaller KL-divergence

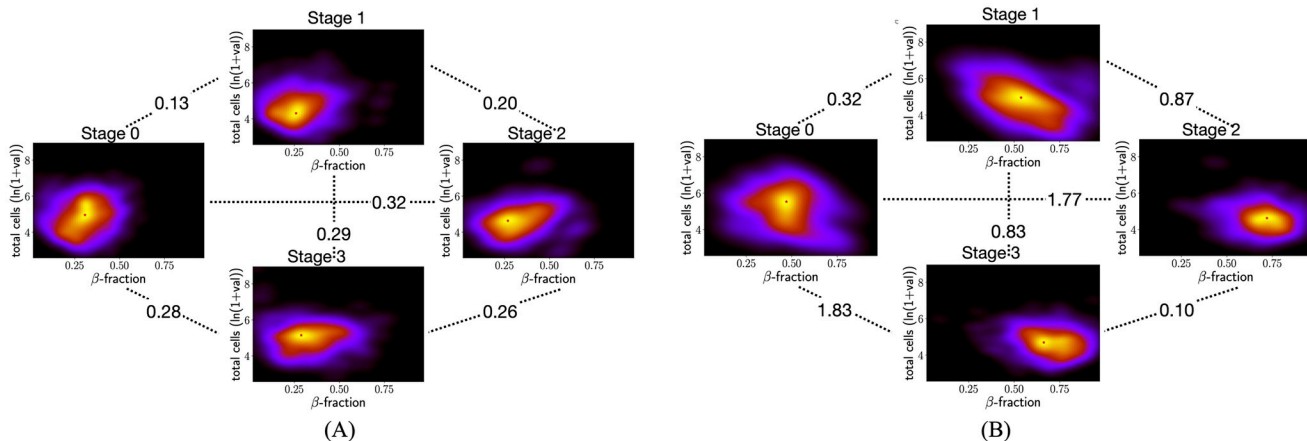

**Fig 4. Comparing KDEs of distributions of islets from developmental data set with (A) at least one NS $\beta$-comp inside an $\alpha\delta$-cycle and (B) at least one NS $\alpha\delta$-comp inside an $\beta$-cycle.** Stages 0 to 3 in clockwise from the left-most plot for stage 0. Peaks in (A) are at (0.31, 4.95), (0.26, 4.31), (0.27, 4.63), and (0.29, 5.14) and in (B) are at (0.47, 5.53), (0.54, 4.95), (0.72, 4.63), and (0.66, 4.69), for stages 0 to 3.

of 0.10 between stages 2 and 3. For such islets, the largest KL-divergence is between stages 0 and 3, which might be attributable to the high KL-divergence between KDEs of all islets in these stages (largest KL-divergence in Fig 2A is between stages 0 and 3). This is supported by the observation that the changes in peaks of KDE are similar in both cases, a larger proportion have higher $\beta$-fraction in stage 3 as compared to stage 0. KS-test estimates significant $p$-values (less than 0.05) for all pairwise comparisons between developmental stages in both cases of islets, with a NS $\beta$-component inside an $\alpha\delta$-cycle and a NS $\alpha\delta$-component inside a $\beta$-component (S2 Table). S10B and S10C Fig show exemplary 2D sections (characteristics similar to the peak of the KDEs) for the developmental stages for islets with at least one $\beta$-component inside a mantle and islets with at least one $\alpha\delta$-component inside a mantle, respectively.

## There are differences in islet cytoarchitecture between control and diabetic subjects

There were 2038 and 1179 islets with at least five $\beta$-cells and five $\alpha\delta$-cells from control and diabetic subjects, respectively. Fig 5A shows that the KL-divergence between KDEs of islets with at least one NS $\beta$-component in an $\alpha\delta$-cycle (middle row in the figure panel) and of those with at least one NS $\alpha\delta$-component in a $\beta$-cycle (bottom row) is more than double the KL-divergence between KDEs of all islets (top row), between control and diabetic subjects. We found 175 and 85 islets with at least one geometric $\alpha\delta$-cycle around a NS $\beta$-component in non-diabetic and diabetic subjects, respectively. 159 and 56 islets were found to have at least one geometric $\beta$-cycle around a NS $\alpha\delta$-component. S2 Table shows that distributions of the islets characteristics between control and diabetic are significantly different in both cases, $p$-values from KS-test are $<0.05$. S11 Fig. shows exemplary 2D sections for each case. Fig 5B left panel shows that percentages of islets with at least one NS component in a cycle are lower for diabetic subjects. A similar trend is observed for maximum dimension-1 persistence (Fig 5B right panel). This correlation between percentages of islets with NS components in cycles and percentiles of the maximum of dimension-1 persistence was also observed across developmental

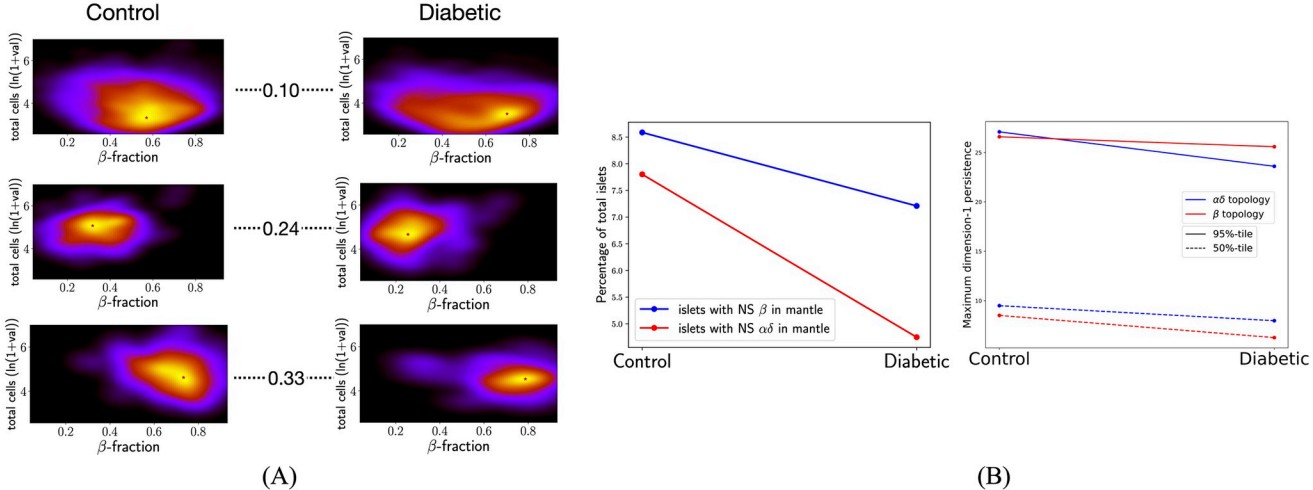

(A)                                                                  (B)

**Fig 5. Comparing features of islets between control and diabetic subjects.** (A) KDEs for control (left column) and diabetic (right column) subjects for all islets (top row), islets with at least one NS $\beta$-component in an $\alpha\delta$-cycle (middle row), and islets with at least one NS $\alpha\delta$-component in a $\beta$-cycle (bottom row). Numbers show the KL-divergence. (B) The percentage of islets that have at least one cycle around a NS component is lower in diabetic subjects as compared to non-diabetic subjects. Percentiles of maximum dimension-1 persistence of islets also are lower for diabetic subjects.

stages. S2 Fig shows that distributions of maximum persistences are significantly different between control and diabetic subjects.

## Cycles are closer to the islet's periphery than its center

Minimal distances of each geometric cycle from its islet's periphery and center were computed. Fig 6A shows that the minimal distance of the computed cycles from the islet's periphery is less than their minimal distance from the islet's center in the T2D data set. Similar was observed for the developmental data set (see S5 Fig). Fig 6B shows that $\alpha\delta$-cycles in small islets (estimated area <10000) are close to the periphery in both control and diabetic. However, there exist $\beta$-cycles in small islets that are far from the periphery, as shown by large minimal distances from the periphery. We also observe that larger islets have some cycles with a larger minimal distance from the periphery. Moreover, only a few cycles (6% to 19%) contain the islet center inside them. S6 Fig shows distance from periphery vs. islet area for the developmental data set.

## All results are consistent between geometric and PH-cycles

For all of the computed geometric cycles, proximal PH-cycles were computed. In at least 89.5% of the islets with at least one geometric cycle around a NS component, a PH-cycle proximal to that geometric cycle was found. Specifically, there were 629, 439, 165, and 440 islets with at least one $\alpha\delta$ PH-cycle around a NS $\beta$-component in stages 0 to 3, respectively. There were 60, 79, 219, and 753 islets with at least one $\beta$ PH-cycle around a NS $\alpha\delta$-component. For the T2D data set, we computed 175 and 85 islets with at least one $\alpha\delta$ PH-cycle around a NS $\beta$-component in non-diabetic and diabetic subjects, respectively. 155 and 54 islets were computed to have at least one $\beta$ PH-cycle around a NS $\alpha\delta$-component. S3 Table shows that

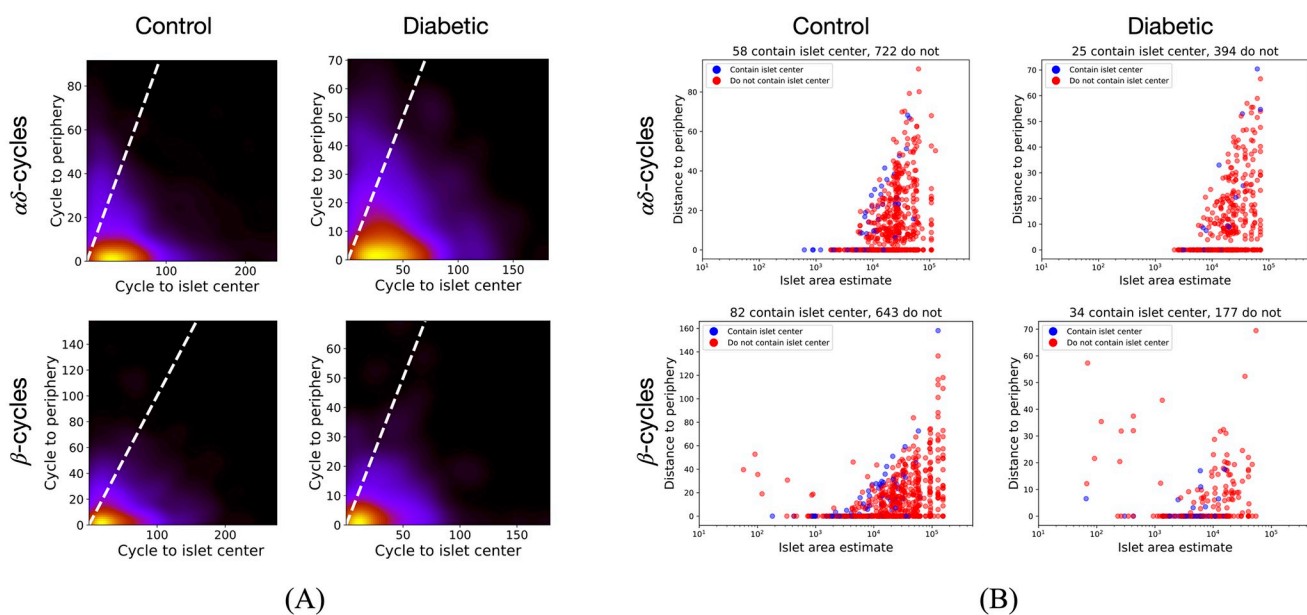

(A)　　　　　　　　　　　　　　　　　　　　　(B)

**Fig 6. Analysis of cycles with respect to islet periphery and center.** (A) KDEs for a minimal distance of cycles from islet periphery vs. islet center. The majority of the cycles are below the $y = x$ line (white dashed) showing that their minimal distance from the periphery is less than that from the islet center. (B) Minimal distances of cycles from islet-periphery vs. islet's estimated area. $\alpha\delta$-cycles in small islets touch the periphery and very few cycles contain the islet center inside them. There are cycles in larger islets that are far from the periphery.

distributions of islet characteristics in both cases are not significantly different for all different categories in both data sets, $p$-values from KS-test are $\gg 0.05$ (at least 0.99). The maximum KL-divergence between KDEs of islets with at least one geometric $\alpha\delta$-cycle around a NS $\beta$-component and those with at least one $\alpha\delta$ PH-cycle around a NS $\beta$-compoment across all stages is 0.001. For islets with at least one $\beta$-cycle around a NS $\alpha\delta$-component, this number is 0.005. For the T2D data set, these numbers are 0 and 0.004. We note that these KL-divergences are significantly smaller (by an order of magnitude) as compared to the divergences observed in previous results. S3 and S4 Figs show all KDEs for developmental and T2D data sets, respectively. KS-tests and comparison of KL-divergences of KDEs give evidence for agreement between results from geometric and PH-cycles.

## PH finds closed polyhedral structures in 3D islets consisting of $\alpha\delta$-cells ($\beta$-cells) around multiple $\beta$-cells ($\alpha\delta$-cells)

We showcase the application of PH to 3D data sets. Structural information of mouse ($n = 29$) and human ($n = 28$) islets were obtained from [52]. Fig 7 illustrates results for three of the islets, two from humans and one from mice.

## Methods

*Data acquisition.* The two data sets for human pancreatic islets in this study comprise of two-dimensional coordinates of beta ($\beta$), alpha ($\alpha$), and delta ($\delta$) cells in islets. The data set with islets at different developmental stages is from human pancreatic tissues that were obtained from the University of Chicago Human Tissue Resource Center with an exemption from the Institutional Review Board [21]. The different stages are gestation (stage 0), 1–35 weeks (stage 1), 12–24 months (stage 2), and 28 months and later (stage 3). The data set with diabetic and non-diabetic human subjects is from [53]. Locations of endocrine cells were obtained as described in the original studies, which we briefly summarize here. Two-dimensional sections of tissue samples were stained for insulin, glucagon, somatostatin, and DAPI. Each section was imaged, and two-dimensional coordinates for each cell nucleus were estimated based on high

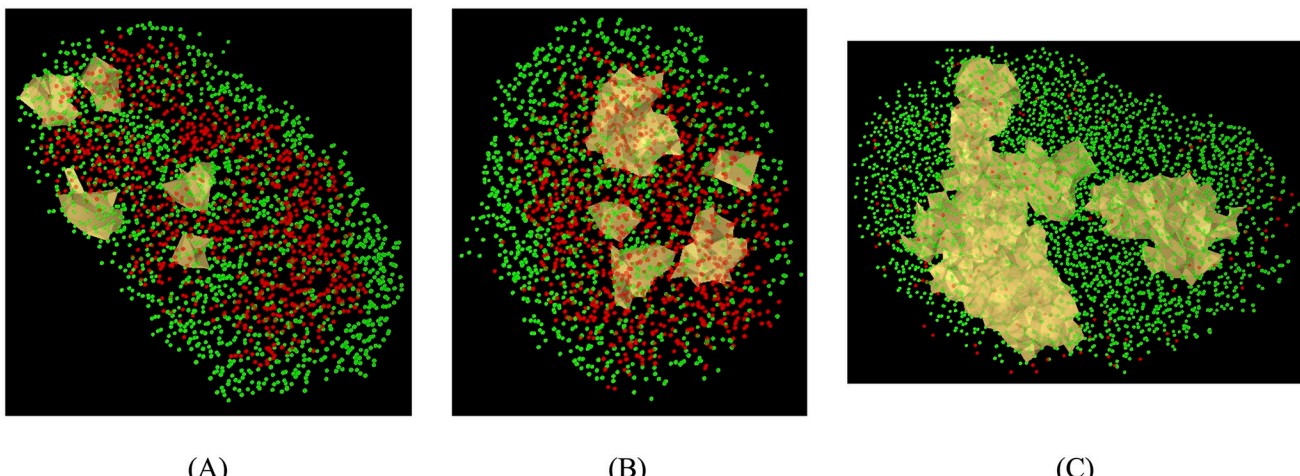

(A)                              (B)                              (C)

**Fig 7. Examples of closed polyhedral structures found by PH in 3D islets.** (A) $\beta$-cycles that contain $\alpha\delta$-cells inside them in a human islet. $\beta$-cells are in green and $\alpha\delta$-cells are in red. (B) $\alpha\delta$-cycles that contain $\beta$-cells inside them in a human islet. (C) $\beta$-cycles thet contain $\alpha\delta$-cells inside them in a mouse islet.

concentrations of DAPI. The cell type of each cell was recorded as $\beta$, $\alpha$, or $\delta$ based on a high concentration of insulin, glucagon, or somatostatin near its nucleus, respectively.

*Defining $\beta$- and $\alpha\delta$-graphs for islets.* Islets with at least five $\alpha\delta$-cells and five $\beta$-cells were considered. $V_{\alpha\delta}^I$ and $V_{\beta}^I$ denote the sets of $\alpha\delta$-cells and $\beta$-cells, respectively, in islet $I$. Edges between $\alpha\delta$-cells and between $\beta$-cells were defined as follows. First, neighborhood radii $\tau_{\beta}^I$ and $\tau_{\alpha\delta}^I$ were computed using the pair distribution function [18]. The pair distribution function is computed as ratio of the number of cells at a radial distance of $r$ to the number of cells expected if they are randomly distributed. The thickness of the radial shells was chosen as 0.5. At $r$-values where peaks occur in the curve of the pair distribution of function, it is expected that there is a larger number of cell pairs within intercell distances than would be expected from a random distribution [21]. The peak at the smallest value of $r$ represents a primary correlation between cells. Peaks of diminishing heights occur at higher $r$-values as a result of secondary correlations between cells. A peak-finding algorithm was implemented to compute $\tau_{\beta}^I$ as the smallest $r$-value at which the pair distribution function for $\beta$-cells is minimal between the second and the third peaks. $\tau_{\alpha\delta}^I$ is computed similarly. The area of an islet was defined as the area of the bounding box around all of its cells. After $\tau_{\beta}^I$ was computed, edges between $\beta$-cells are initially all the $\beta$-cells that are at most with $\tau_{\beta}^I$ distance apart. Edges between $\alpha\delta$-cells were initialized similarly. Second, a shadow algorithm was implemented to account for obstruction in the interaction of two cells due to the presence of a third cell between them [18]. All edges that are obstructed by a cell were removed. The final sets of $\beta$-edges and $\alpha\delta$-edges are denoted by $E_{\beta}^I$ and $E_{\alpha\delta}^I$, respectively.

*Computing cycles using geometry.* $\alpha\delta$-cycles around $\beta$-cells were computed as follows. The $\alpha\delta$-graph of islet $I$ is the discrete graph on its $\alpha\delta$-cells, denoted by $G_{\alpha\delta}^I \equiv (V_{\alpha\delta}^I, E_{\alpha\delta}^I)$. $G_{\beta}^I$ is defined similarly. We drop the superscript $I$ for notational convenience. A $\alpha\delta$-cycle is a simple closed curve on $G_{\alpha\delta}$ such that it partitions the graph into two disjoint sets, one inside the cycle and one outside. It follows from the Jordan Curve Theorem that the graph has to be planar. $G_{\alpha\delta}$ was made planar using dummy vertices—if two edges, $\{v_1, v_2\}$ and $\{v_3, v_4\}$ intersect at $p$, then they are removed from $E_{\alpha\delta}^I$, a dummy vertex $u_p$ located at $p$ is added to $V_{\alpha\delta}$, and edges $\{v_i, u_p\}$, $1 \leq i \leq 4$, are added to $E_{\alpha\delta}^I$. Geometric $\alpha\delta$-cycles around $\beta$-cells were computed using three main steps. First, a list of all $\alpha\delta$-cycles was computed as follows. For each connected component of $c_k$ of $G_{\alpha\delta}$, a spanning tree $T_k$ is constructed on $c_k$. Let $F_k$ be the set of edges that are in $E_{\alpha\delta}$ but not in $T_k$. The weight of an edge between two cells is defined as the Euclidean distance between the cells. For each edge $\{v_i, v_j\}$ in $F_k$ a shortest weighted path $P$ between $v_i$ and $v_j$ in $T_k$ is computed. The set of edges $\{\{v_i, v_j\}\} \cup \{\text{edges in } P\}$ forms an $\alpha\delta$-cycle or a cycle of $\alpha\delta$-cells in $c_k$. $\alpha\delta$-cycles in all components of $G_{\alpha\delta}$ were computed. Second, $\alpha\delta$-cycles that surround $\beta$-cells were determined. For each $\beta$-cell, all $\alpha\delta$-cycles that contain it are computed using a winding-number algorithm. A set of $\beta$-cells can be inside multiple cycles. Sets of $\beta$-cells were computed that are inside the same set of $\alpha\delta$-cycles. Let $\mathcal{B}$ be the collection of such sets of $\beta$-cells. Third, for each set $S$ of $\beta$-cells in $\mathcal{B}$, a minimal geometric cycle, $P_S$, of $\alpha\delta$-cells is defined and computed as follows.

1. Determine the closest pair of cells $\{\beta_i, v_*\}$ for $\beta_i \in S$ and $v_* \in V_{\alpha\delta}$. Initialize $P_S = [v_*]$. The aim is to find path $P_S = [v_*, v_1, \ldots, v_n, v_*]$ of $\alpha\delta$-cells that contains $S$ inside it.

2. Let $\text{ang}(v, u) = \arctan \frac{v_y - u_y}{v_x - u_x}$, where $v_x$ and $v_y$ are $x$ and $y$-coordinates of vertex $v$ in the islet. If $\text{ang}(v, u) < 0$, then define $\text{ang}(v, u) = \arctan \frac{v_y - u_y}{v_x - u_x} + 2\pi$. In other words, it is the positive or counter-clockwise angle that the horizontal line through $u$ has to turn to be parallel to the line through $u$ and $v$. Let $\theta_* = \text{ang}(\beta_i, v_*)$.

3. For $v_j \in V_{\alpha\delta}$, let $N(v)$ be the neighbors of $v_j$ in $G_{\alpha\delta}$. Let angles$(v_j) = [\text{ang}(v_j^m, v_j)] = [\theta_m]_{1 \leq m \leq |N(v_j)|}$ be the sorted list of angles in increasing order, where $v_j^m \in N(v_j)$.

4. The notion of minimality that we define is that the counterclockwise turn at every node along the cycle should be minimal.

5. Hence, $v_1 \in P_S$ is computed as $v_1 = v_*^k \in N(v_*)$ such that $\theta_k$ is the smallest angle greater than $\theta_*$. Once we have $P_S = [v_*, v_1]$, the next cells in the path are computed as follows. If $v_m$ and $v_{m+1}$ are two adjacent cells in $P_S$, the next cell $v_{m+2} \in P_S$ is computed as $v_{m+2} = v_{m+1}^k \in N(v_{m+1})$ such that $\theta_k$ is the smallest angle greater than ang$(v_m, v_{m+1})$. The next cells are computed till we reach $v_*$. Note that if no such $\theta_k$ exists at a step, then the minimal counterclockwise turn is for $k = 1$, hence, $v_{m+2} = v_{m+1}^1$.

6. It is possible that the computed minimal path does not contain $S$ inside it. Hence, we check that every $\beta$ cell in $S$ is inside the computed path using the winding number algorithm. If not true, then the pair $\{\beta_i, v_*\}$ is marked as an incompatible pair and we begin with step 1 by finding the closest pair but ignoring the ones that are marked as incompatible.

Sizes of components of $S$ are determined using the Networkx Python package [54]. $\beta$-cycles around $\alpha\delta$-cells are computed similarly.

An advantage of using angles to find geometric loops is that we can identify components that are partially surrounded by cells of the other kind (see S7 Fig). However, analysis of partial loops is not included in this work.

*Kernel density estimation and Kullback-Leibler divergence.* Islets are characterized by $\beta$-cell fraction and the total number of cells in them. The number of cells in islets was transformed by $\ln(1 + x)$. KDE was estimated using `scipy.stats.gaussian_kde` module of the Scipy Python package [55], with the default method for bandwidth estimation. Two-dimensional KDE was computed on a grid of resolution 100 over the space of ($\beta$-fraction, the number of total cells in islets) coordinate pairs. KL-divergence between two KDEs was computed using `scipy.stats.entropy` module with default settings.

*Computing islet's periphery and distance of cycle from its islet's periphery and center.* Since the periphery of a 2D slice can be non-convex, it was estimated by computing an alpha shape [56] for the set of all cells in the slice as follows. The alpha shape computation depends on the hyperparameter called the shrink factor. Shrink factor set to 0 computes the convex hull of the set of points as the alpha shape. To get a more accurate estimate of the non-convex periphery, we first initialized the shrink factor as the multiplicative inverse of the maximum of $g_\beta^I$ and $g_{\alpha\delta}^I$ for islet $I$. If the computed alpha shape was composed of multiple polygons, the shrink factor was halved and the alpha shape was computed again. The periphery of the islet was defined by the alpha shape that was composed of a single polygon at the largest possible shrink factor in this iteration. The computation was done using the Python package `alphashape v1.3.1`. The area of the islets and distances between cycles and the periphery were estimated using the `distance` method from Python package `Shapely v2.0.1`. `distance` computes the distance between two polygons as the distance between the closest pair of points. The islet center was computed as the centroid of the periphery. Containment of the islet center inside a cycle was computed using `contains` method of `Shapely v2.0.1`.

*Computing dimension-1 persistent homology.* Dimension-1 PH of $\alpha\delta$-cells was computed using the standard column algorithm to reduce boundary matrices [57]. An introduction to persistent homology with precise mathematical terminology can be found at [58]. Here we provide a brief overview of the standard column algorithm to compute dimension-1 persistence pairs using non-technical terminology that might be more accessible to non-experts.

Total ordered sets of vertices, edges, and triangles are defined as follows. Vertices in $V_{\alpha\delta}$ are indexed aribitrarily. All possible edges on $V_{\alpha\delta}$ are indexed by their length with longer edges having a higher index. All possible triangles on $V_{\alpha\delta}$ are indexed by order of the edge with the largest order in their boundary. In both cases, ties are broken arbitrarily. Boundary matrix for edges, $D_e$, is defined as a $m$ by $n$ matrix with, $D_e[i, j] = 1$ iff $v_i$ is in the boundary of $e_j$ and $D_e[i, j] = 0$, otherwise. Similarly, boundary matrix for triangles, $D_t$, is defined as a $n$ by $k$ matrix with, $D_t[i, j] = 1$ iff $e_i$ is in the boundary of $t_j$ and $D_t[i, j] = 0$, otherwise. Boundary matrices are reduced using standard column reduction as follows. A column is non-empty if it has at least one non-zero entry. Pivot-index of a non-empty column is defined as the maximum row index with a non-zero element. If columns $i$ and $j$ have the same pivot-index and $i < j$, then column $i$ is replaced with its modulo 2 sum with column $j$. This is repeated till no two non-empty columns have the same pivot-index. $D_e$ and $D_t$ are reduced independently, and the resulting reduced matrices are denoted by $R_e$ and $R_t$. The reduction operations are denoted by $V_e$ and $V_t$, respectively. If $(i, j)$ is a pivot of $R_t$, then there is a topological feature born at the spatial scale of the length of edge $e_i$ and it dies at the spatial scale of the largest length of the edge in the boundary of triangle $t_j$. Persistence of each topological feature is the difference between its death and birth. Dimension-1 PH of $\beta$-cells was computed similarly.

*Computing an initial set of biologically significant cycles using persistent homology.* We provide instructions to compute representative boundaries using non-technical language. See [59] for an explanation of the algorithm using precise terminology. $\alpha\delta$-cycles containing $\beta$-cell(s) and $\beta$-cycles containing $\alpha\delta$-cells(s) are classified as biologically significant. To compute $\alpha\delta$-cycles around $\beta$-cells in an islet $I$, sets of vertices, edges, and triangles were defined as follows. Vertices and edges are $V_{\alpha\delta}^I$ and $E_{\alpha\delta}^I$, respectively. Triangles are those that have edges in $E_{\alpha\delta}^I$ and do not contain (horizontal ray algorithm) any $\beta$-cell. Boundary matrices are defined as described previously and PH is computed for this collection of vertices, edges, and triangles. Since triangles containing $\beta$-cells are not in the boundary matrix, topological features in the $\alpha\delta$-graph that contain $\beta$-cells will not die. If column $i$ of $R_e$ is empty and $i$ is not a pivot-index of any column of $R_t$, then column $i$ of $V_e$ is a representative boundary of a topological feature that does not die. From these representative boundaries we ignore the ones that do not contain any $\beta$-cells inside them. This results in an initial set of $\alpha\delta$-cycles that contain at least one $\beta$-cell inside them.

*Greedy and stochastic shortening of PH-cycles before comparison with geometric cycles.* Representative boundaries around topological features are not unique by definition and can be geometrically imprecise. To improve their precision before comparison with geometric cycles, the boundaries in the initial set were shortened using greedy and stochastic shortening introduced in previous work [51]. Technical details of stochastic shortening of $\alpha\delta$-cycles in an islet are as follows. Locations of $\alpha\delta$-cells were perturbed 50 times in neighborhood disks centered at the cells. Edge-lengths were rounded to the nearest integer. Since edges of the same length can be ordered arbitrarily, at most 50 unique different total ordered sets of edges were constructed for each perturbation. PH-cycles for each permutation of every perturbation were computed as described above. Moreover, this was done for ten different values of maximum neighborhood disk radii of $[0.1, 0.2, \ldots, 1]$. These ranges of values were chosen because they are much less than ($\approx 10\%$) the minimum neighborhood radius of 8 that was computed across all islets in both data sets. This resulted in up to 25000 sets of representative boundaries for the islet. For each boundary in a set of representatives, the set of $\beta$-cells inside it was computed. For each set of $\beta$-cells that is inside some representative boundary, a list of those with the least number of edges was constructed. Finally, we computed if any of the representative boundaries in this list is proximal to a geometric cycle as described next.

*Comparing geometric cycles with PH-cycles.* We say a geometric $\alpha\delta$-cycle matches an $\alpha\delta$ PH-cycle if they both contain the same set of $\beta$-cells inside them. Suppose $L$ is the set of $\alpha\delta$-cells that are computed to form a geometric cycle and $\hat{L}$ is an $\alpha\delta$ PH-cycle computed for an islet $I$. The distance between them is defined as $d(L, \hat{L}) =$

$\max\{\max_{p_i \in L}\{\min_{\hat{p}_j \in \hat{L}}\{d(p_i, \hat{p}_j)\}\}, \max_{\hat{p}_i \in \hat{L}}\{\min_{p_j \in L}\{d(p_i, \hat{p}_j)\}\}\}$, where $d(p, q)$ is the Euclidean distance between cells $p$ and $q$ in the islet. We say that $L$ and $\hat{L}$ are proximal if cycle $L$ matches with $\hat{L}$ and $d(L, \hat{L}) \leq \tau_{\alpha\delta}^I$. Otherwise, we say they are distant. Analogous definitions follow for proximal $\beta$ geometric and PH-cycles.

*Computing cycles in 3D data sets using PH.* To compute $\alpha\delta$-cycles around $\beta$-cells in an islet $I$, sets of vertices, edges, and triangles were defined as before. Additionally, tetrahedrons on $\alpha\delta$-cells are defined as those that have edges in $E_{\alpha\delta}^I$, all faces as valid triangles, and do not contain any $\beta$-cell. Containment was checked using barycentric coordinates. Tetrahedrons are ordered by the length of the longest of the edges of their faces, also called their diameters. Those with the same diameter are given a unique order arbitrarily. This results in a full-ordered set of tetrahedrons. The boundary matrix for tetrahedrons is defined and constructed, denoted by $D_h$. It is reduced as before to give the reduced matrix $R_h$ and features that do not die are computed using methods analogous to those defined for the 2D case. The threshold to define edges on the graph was chosen as 25.

*Tests for statistical significance.* 1D distributions of maximum persistences were compared using two-sided Mann-Whitney U rank test for two independent samples using `scipy.stats.mannwhitneyu` module with default settings. 2D distributions of islet characteristics were compared by computing *p*-value from KS-test using the `ndtest` Python package from https://github.com/syrte/ndtest.

## Discussion

Studies of the structure of islets of Langerhans have shown that the relative number and arrangement of the individual cell types plays a critical role in regulating glucose metabolism [60]. The arrangement is highly complex and heterogeneous, and has been investigated using various experimental and quantitative approaches, including network science methods [41, 42]. Changes in the structural characteristics of islet cell types have been observed during the progression of type 2 diabetes, with decreased beta-cell numbers and disrupted structural and functional connectivity being key features of the disease. Quantification of islet structure has typically been applied to 2D images of islet sections, but such 2D data is unable to capture important aspects of islet physiology such as vasculature [61] and innervation [62], both of which are known to play critical roles in islet function and in functional communication between islets.

Our contribution here is two-fold. We have developed two distinct approaches to go beyond functional network statistics or spatial descriptive or network statistics in the characterization of islet cytoarchitecture. One, a geometric approach, is much easier to apply to, and visualize in, 2D image data, and the other, a topological characterization, is applicable to 2D and 3D data. Of note here is that in contrast to network characterizations, the topological features we uncover are nonlocal by construction, and therefore are capturing a complementary view of islet cytoarchitecture relative to network approaches. While there are other computational approaches to topological characterization, our approach is the only one that explicitly gives the locations of the actual topological features in the image. Such location information is, of course, the sine qua non for studies of the functional impact of any feature, see, for example

[63]. We confirmed that results from our two distinct computational methods, geometric and topological, completely agree for the 2D sections for the developmental and T2D data sets.

Our results showed that a low percentage of islets contained ring structures with a NS-component of the other kind of cells. However, we observe changes in this percentage across developmental stages and between control and diabetic, that correlate with trends in persistence homology of the islets in both cases. These differences in islet cytoarchitecture may affect paracrine signaling between endocrine cells resulting in functional differences [7, 10–13]. For example, NS-components of $\beta$-cells might be indicative of $\beta$-cells coupled via gap junction linkages that may play a functional role in coordinated responses to endogenous insulin secretagogues such as glucagon-like peptide-1 (GLP-1), but might not be significant for islet dynamics involved in glucose-stimulated $Ca^{2+}$ oscillations [64, 65]. Further, 3D structural analysis of human islets has shown that $\alpha$-cells are arranged along interiorly pervading vessels [7]. Hence, studying the topology of the 3D islets taking into account the blood vessel information, and comparing it between control and diabetic subjects might be important to study possible relations between morphological and functional changes.

The topological characterization [66] may be important for understanding disease susceptibility of islets of different characteristics. It is obvious that 2D images provide a limited view of the complex 3D architecture of the islet, and can result in under- or overestimation of cell sizes and numbers. In addition, automated segmentation algorithms may not always accurately distinguish individual cells, particularly in cases where cells are tightly packed or have irregular shapes. Finally, variations in staining or imaging conditions can affect the accuracy and reproducibility of quantitative measurements. The topological characterization is robust to many of these experimental uncertainties.

3D imaging techniques, including confocal microscopy, two-photon microscopy, and optical coherence tomography, are being developed to provide a more comprehensive understanding of islet architecture [19, 43, 61, 67–73], including progress on visualization of vasculature. Advances in image analysis algorithms allow the segmentation of individual cells from 3D image stacks, allowing for the quantification of various structural and functional parameters. Our topological approach can be applied without any changes to 3D imaging data, while the geometric approach is difficult to generalize to 3D without mathematical assumptions. However, current 3D imaging of islets does have limitations, such as limited penetration depth and imaging speed, which may result in incomplete imaging of large islets. Nevertheless, we illustrated application of PH to find $\alpha\delta$-cycles (closed polyhedral structures) in 3D data sets containing $\beta$-cells inside (and $\beta$-cycles containing $\alpha\delta$-cells inside) for the limited data sets of human and mice that were publicly available. It can be of interest to analyze the properties of locations of the cycles found in 3D data sets. However, in the limited 3D data sets available, we observed islets with highly convex shapes and some with multiple globules. Mathematically sound definition and stable computation of geometrical properties of these 3D point-clouds, for example equatorial plane and poles, might require a larger number of data sets for testing and validation. It can be of further interest to compare the consistency of results between 3D data sets and their 2D sections for different slicing schemes. These analyses of 3D islet cytoarchitecture can be a future direction of research as data from many 3D islets becomes available.

Mathematical models have been used to simulate the behavior of islets and predict how changes in cell number, size, and arrangement will affect glucose metabolism. These models can also be used to analyze the effects of different interventions, such as drug treatments, on islet function. Applying this type of mathematical modeling [37, 47, 48, 74] to simulated islet cell distributions with similar network characteristics but distinct topological characteristics or

vice versa would be an interesting future direction to determine the relevance of nonlocal topological features to islet function.

The contribution of this work to biology is to provide quantitation of structures that have been controversial in terms of existence and/or functional significance. The results are correlative and not causative. How these structures are related to glucose-stimulated insulin secretion (GSIS) profiles is unclear. We hope that by providing multiple mathematical methods for defining and computing such topological structures, the field can focus on the relevance of these structures for function and understand how different features in the GSIS profile are related to specific islet features.

## Supporting information

**S1 Appendix. Persistent homology applied to discrete set of points.** An illustration of dimension-1 PH computed for a point-cloud and intuitive interpretation of the results.
(PDF)

**S1 Table. KS-test $p$-values for pairwise comparison of distributions of islet characteristics for all islets.**
(PDF)

**S2 Table. KS-test $p$-values for pairwise comparison of distributions of islet characteristics for islets with cycles around NS components.** Upper diagonal matrix shows comparison for islets with a NS $\beta$-component in an $\alpha\delta$-cycle. Lower diagonal matrix shows comparison for islets with a NS $\alpha\delta$-component in a $\beta$-cycle.
(PDF)

**S3 Table. KS-test $p$-values for comparison between geometric and PH-cycles, of distributions of islet characteristics for islets with cycles around NS components.**
(PDF)

**S4 Table. Demographics of the subjects in the T2D data set.**
(PDF)

**S1 Fig. Comparing distributions of maximum dimension-1 persistence between developmental stages.** Top row is $\alpha\delta$-cells. Bottom row is $\beta$-cells. Left column shows box plots. Right column shows the significance results from pairwise Mann-Whitney U tests. Black dotted edges represent $p$-value $>0.05$ and solid thick red edges represent $p$-value $<0.001$. Only stages 0 and 1 are not significantly different.
(TIF)

**S2 Fig. Comparing distributions of maximum dimension-1 persistence between control (C) and diabetic (D) subjects.** Left column is for topology of $\alpha\delta$-cells and right column is for topology of $\beta$-cells. Both are significantly different since $p$-value $<0.05$.
(TIF)

**S3 Fig. KDEs of features of islets with cycles around NS components are similar for geometric and PH-cycles for developmental data set.** KDEs of islets with (A) $\alpha\delta$-cycles around at least one NS $\beta$-component, and (B) $\beta$-cycles around at least one NS $\alpha\delta$-component. In each panel, left column is for geometric cycles and right column is PH-cycles. Rows are stages are 0 to 3 from top to bottom.
(TIF)

**S4 Fig. KDEs of features of islets with cycles around NS components are similar for geometric and PH-cycles for T2D data set.** KDEs of islets with (A) $\alpha\delta$-cycles around at least one

NS $\beta$-component, and (B) $\beta$-cycles around at least one NS $\alpha\delta$-component. In each panel, left column is for geometric cycles and right column is PH-cycles. Top row is non-diabetic or control and bottom row is for diabetic subjects.
(TIF)

**S5 Fig. Peripheral analysis for developmental data set.**
(TIF)

**S6 Fig. Distance of mantles from periphery vs. estimate of the islet area.**
(TIF)

**S7 Fig. Partial loop computed using geometric method.** A component of $\beta$-cells (green) is partially surrounded by $\alpha\delta$-cells (small red points). The red arcs around the $\beta$-cells show the region surrounded and the black arcs show the region not surrounded.
(TIF)

**S8 Fig. Comparing distributions of maximum dimension-1 persistences for developmental data set.** Except for developmental stages 0 and 1, all other pairwise comparisons showed significant difference ($p \ll 0.05$) using Mann-Whitney U test.
(TIF)

**S9 Fig. Characterizing ratio of $\alpha$ to $\delta$ cells in 2D sections for the T2D data set.**
(TIF)

**S10 Fig. Exemplary 2D sections for developmental data set.** Examples of sections with characteristics close to the peak in KDE for (A) all islets, (B) islets with at least one NS $\beta$-component surrounded by a cycle, and (C) islets with at least one NS $\alpha\delta$-component surrounded by a cycle.
(TIF)

**S11 Fig. Exemplary 2D sections for the T2D data set.**
(TIF)

## Acknowledgments

We thank Dr. Wolfgang Resch for helping us in utilizing the computational resources of the NIH HPC Biowulf cluster (http://hpc.nih.gov).

## Author Contributions

**Conceptualization:** Manu Aggarwal, Deborah A. Striegel, Manami Hara, Vipul Periwal.

**Data curation:** Manami Hara.

**Formal analysis:** Manu Aggarwal, Deborah A. Striegel.

**Funding acquisition:** Vipul Periwal.

**Investigation:** Manu Aggarwal, Deborah A. Striegel.

**Methodology:** Manu Aggarwal, Deborah A. Striegel, Vipul Periwal.

**Project administration:** Vipul Periwal.

**Resources:** Vipul Periwal.

**Software:** Manu Aggarwal, Deborah A. Striegel.

**Supervision:** Vipul Periwal.

**Visualization:** Manu Aggarwal.

**Writing – original draft:** Manu Aggarwal, Vipul Periwal.

**Writing – review & editing:** Manu Aggarwal, Deborah A. Striegel, Manami Hara, Vipul Periwal.

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
