## [Decision Letter · Decision Letter 0]

21 Jul 2023

Dear Dr. Aggarwal,

Thank you very much for submitting your manuscript "GEOMETRIC AND TOPOLOGICAL CHARACTERIZATION OF THE CYTOARCHITECTURE OF ISLETS OF LANGERHANS" for consideration at PLOS Computational Biology.

As with all papers reviewed by the journal, your manuscript was reviewed by members of the editorial board and by several independent reviewers. In light of the reviews (below this email), we would like to invite the resubmission of a significantly-revised version that takes into account the reviewers' comments.

We cannot make any decision about publication until we have seen the revised manuscript and your response to the reviewers' comments. Your revised manuscript is also likely to be sent to reviewers for further evaluation.

Sincerely,

Philip K Maini

Academic Editor

PLOS Computational Biology

Pedro Mendes

Section Editor

PLOS Computational Biology

Reviewer's Responses to Questions

**Comments to the Authors:**

Reviewer #1: Review is attached.

Reviewer #2: This article takes a dual geometrical and then topological view of islet architecture to consider the impact of non-local cellular interactions (e.g. cycles of delta cells around beta cells, spherical enclosure of beta cells around alpha cells).

The authors extend their previous work on islet graph analysis to account for mantle identification. They describe in good detail the algorithm used and inspire confidence in their method.

They show a significant disruption in the mantle structure of type 2 diabetes islets, which agrees with previous results using other measures.

I think this is a valuable and technically sophisticated extension of topological methods to islet structure/function discussions.

There are a few questions, suggestions, and minor edits:

Code to generate the results (or similar) should be published for reproducibility.

Topologies that are considered appear to be fundamentally two-dimensional which may not always be an appropriate reduction (e.g. untrapped from a cycle in a third dimension). While the ability to discuss 3D structure is clear in the method, the authors do not seem to venture an example where 3D data is involved? Can such an example, say from slice (or other) data, be included to support the applicability in 3D?

Can you confirm enclosure of NS components (i.e. while centers of cells may appear to enclose components perhaps the boundary could be breached by an oddly shaped neighboring cell)? Are there examples of data with cell membrane markers included to tesselate the space and confirm mantle enclosure that is suggested by cell centers? This also relates to how "Edges obstructed by a cell are removed"(pg 7) Is there a certain tolerance for edge removal(i.e. if the obstructing cell center is within 1-5 microns along say the orthogonal projection to the edge, then the edge is removed)?

Is it surprising that so low (649/6088, 463/4942, 168/3130, and 453/7203) of fractions of islets at the four stages (assuming it is the same data sets/numbers shown in Fig 3) had an alphadelta-mantle with NS beta-component? Around 10% (or less for later stages)? Does this undermine the importance of these structures in islet function? Or are there other features that might act to prevent connection of mantles in the calculation but yet yielding de facto mantle-like function (e.g. Blood vessels, etc) and increasing the effective mantle structures in reality?

An aggregate analysis made up of the known stages (Figs 2-4) might be useful to then connect with the control and T2D data where you are without stage information. For example, does the control data look like the combination of the data from Figs 2-4? If not, would you be able to predict/suggest what stage (or combination of stages) the control data most resembles?

Is it useful to apply your analysis to an example with mice or rat data data with more straightforward core and mantle geometry?

The authors mention stochastic shortening in mantle calculation of PH-mantles, but not clear where this was utilized in the calculations.

Minor edits:

The authors mention insulin and glucagon in first part of abstract, but then mention delta cells in geometry. Is somatostatin an important regulator or is the electrical connections with delta cells more important? Alpha and delta cells are merged in the analysis, but a more parallel introduction would read better.

When implementing the dirac delta function as a practical matter, is there a tolerance (i.e. cells with a distance of r +/- 1% of r are counted as being at a distance r away)? (pg 7)

Geometry subscript on beta-graph is denoted by alphadelta rather than beta. (pg 7)

Adding a "Stage x" (x being 0, 1, 2, or 3) in white in the upper right corner of each of the four graphs in Fig 2 (and similar latter graphs) would help. Even though the authors were explicit in the caption where to start stage 0 I still took it as the positive x-axis (East rather than West) to start.

Labeling the columns (control vs. diabetic) in Fig 5 would support ease of comparison.

Reviewer #3: Please find below my minor and major comments organized in a systematic manner. I think your article is potentially of great interest to the beta cell community and other fields dealing with similar problems of quantifying architectural features in a robust manner on large data sets. However, I think the article could benefit a lot by taking into account my suggestions and comments.

Minor

Introduction

1-2 % of islet mass may be a slight underestimation, please see Dolensek et al Islets 2015 for more data.

Islets secrete hormones that are crucial for regulating blood glucose levels, as well as levels of amino acids, free fatty acids, keto acids, glycerol, and other energy-rich nutrients. Given the often too glucocentric view on islets and diabetes, perhaps glucose could be replaced by energy-rich nutrients.

In second paragraph of the introduction, the influence of insulin on glucose is pointed out, but its effects extend beyond that, to storage of glucose in the liver in the form of glycogen, as well as to promoting storage of lipids in fat tissue (inhibition of the hormone-sensitive lipase, promotion of lipoprotein lipase), and amino acids in muscle tissue.

In the second or third paragraph of the introduction, it may be worth mentioning that the relationship between islets and acinar and ductal cells may also be different at different developmental stages and in different species (e.g., islets are more intralobular in humans and more interlobular in mice).

In paragraph 4, I think it is important to mention that early during development of diabetes there may be an adaptive increase in mass or function (or both) and that in some people (non-progressors), this may be more functional or persistent than in others (progressors) and is crucial for our understanding why some people do not develop T2D despite insulin resistance and why some respond to dietary interventions while others don´t. See for instance Taylor et al Cell Metab 2018, Stozer et al Nephron 2019, Boland et al Mol Metab 2017, Cohrs et al Cell Rep 2020, Hudish et al JCI 2019, Weir&Weir Diabetes 2004.

2 Results

Has the developmental data set or part thereof been used before in previous publications? If yes, please specify.

What were the donor characteristics for control and T2D human islets and for developmental donors? If available, a table summarizing the main characteristics would be much appreciated. If not, please point out that the data are not available.

Why do the authors combine alpha and delta cells into the alpha-delta group? How does this affect the results?

Please explain the abbreviation NS components.

Figures 2, 4, and 5 would be easier to read with stages (T2D status) indicated also on figures (there is enough space in the top black section to add stage 0, stage 1 etc. Additionally, for the more old-fashioned beta cell physiologists, some exemplary/representative 2D sections for each stage (and CTRL vs T2D) next to KDE plots would be much appreciated.

Discussion

Functional connectivity seems to be reduced under diabetogenic conditions (islets cultured in high glucose with or without palmitate, islets incubated in cytokines, islets from T2D donors) and this has been explicitly or implicitly attributed to decreased gap-junctional connectivity or disrupted cytoarchitecture (e.g., vessels disrupting beta cell syncytium). See for instance work by Richard Benninger, Guy Rutter, David Hodson, and Andraz Stozer. Do your findings of less mantles in T2D islets comply with these findings? Does the lower number of mantles in any way support the view that the beta cell syncytium may be less well connected, that there may be more bottlenecks for intercellular waves spreading between cells? Please include a short paragraph on the relationship between your morphological and these functional findings.

Major

Are for any of the data sets 3D stacks of 2 D sections available? If yes, it would be a great added value to show to what extent the computed/determined characteristics are stable or heterogeneous within islets, i.e., from one pole, over the equatorial plane, and to the other pole. I reckon that such analysis would also enable the authors to determine whether the mantles can be found at all parts of an islet or preferentially closer to the poles or the equatorial plane. I understand that such sections may not be available. In that case, the authors could briefly point this out as a drawback and suggest some ideas for further studies and analyses.

Additionally, for the existing 2D data set, I think it would be of great value to include an analysis of the positions of the mantles found in islets with respect to islet center/periphery.

**Have the authors made all data and (if applicable) computational code underlying the findings in their manuscript fully available?**

Reviewer #1: Yes

Reviewer #2: **No: **

Reviewer #3: Yes

PLOS authors have the option to publish the peer review history of their article (what does this mean?). If published, this will include your full peer review and any attached files.

Reviewer #1: No

Reviewer #2: No

Reviewer #3: **Yes: **Andraž Stožer
---

## [Decision Letter · Decision Letter 1]

19 Oct 2023

Dear Dr. Aggarwal,

We are pleased to inform you that your manuscript 'GEOMETRIC AND TOPOLOGICAL CHARACTERIZATION OF THE CYTOARCHITECTURE OF ISLETS OF LANGERHANS' has been provisionally accepted for publication in PLOS Computational Biology.

Best regards,

Philip K Maini

Academic Editor

PLOS Computational Biology

Pedro Mendes

Section Editor

PLOS Computational Biology

Reviewer's Responses to Questions

**Comments to the Authors:**

Reviewer #1: Your revision has addressed the largest issues raised in my original review.

Reviewer #2: The authors have extensively revised their manuscript in response to reviews as to improve the article considerably. I especially appreciate your inclusion of Figure 7 with 3D application that I feel greatly enhances the potential impact of these methods. Though I have not taken advantage of it, I also appreciate the extensive dissemination of code on their github site. I am satisfied with the revision.

Reviewer #3: Dear Authors,

I greatly value your effort taking into account all of my suggestions and comments to the greatest possible extent and look forward to seeing your work published, recommending it to colleagues in the field, and referring to it in our own future work.

**Have the authors made all data and (if applicable) computational code underlying the findings in their manuscript fully available?**

Reviewer #1: Yes

Reviewer #2: Yes

Reviewer #3: Yes

PLOS authors have the option to publish the peer review history of their article (what does this mean?). If published, this will include your full peer review and any attached files.

Reviewer #1: No

Reviewer #2: No

Reviewer #3: **Yes: **Andraž Stožer

---

## [Editor Report · Acceptance letter]

6 Nov 2023

PCOMPBIOL-D-23-00756R1 

GEOMETRIC AND TOPOLOGICAL CHARACTERIZATION OF THE CYTOARCHITECTURE OF ISLETS OF LANGERHANS

Dear Dr Aggarwal,

I am pleased to inform you that your manuscript has been formally accepted for publication in PLOS Computational Biology. Your manuscript is now with our production department and you will be notified of the publication date in due course.

With kind regards,

Zsofia Freund
